# A Non-Functional Carbon Dioxide-Mediated Post-Translational Modification on Nucleoside Diphosphate Kinase of *Arabidopsis thaliana*

**DOI:** 10.3390/ijms25020898

**Published:** 2024-01-11

**Authors:** Harry G. Gannon, Amber Riaz-Bradley, Martin J. Cann

**Affiliations:** 1Department of Biosciences, Durham University, South Road, Durham DH1 3LE, UK; harry.g.gannon@durham.ac.uk (H.G.G.);; 2Biophysical Sciences Institute, Durham University, South Road, Durham DH1 3LE, UK

**Keywords:** carbon dioxide, post-translational modification, carbamate, *Arabidopsis*, nucleoside-diphosphate kinase

## Abstract

The carbamate post-translational modification (PTM), formed by the nucleophilic attack of carbon dioxide by a dissociated lysine epsilon-amino group, is proposed as a widespread mechanism for sensing this biologically important bioactive gas. Here, we demonstrate the discovery and in vitro characterization of a carbamate PTM on K9 of *Arabidopsis* nucleoside diphosphate kinase (*At*NDK1). We demonstrate that altered side chain reactivity at K9 is deleterious for *At*NDK1 structure and catalytic function, but that CO_2_ does not impact catalysis. We show that nucleotide substrate removes CO_2_ from *At*NDK1, and the carbamate PTM is functionless within the detection limits of our experiments. The *At*NDK1 K9 PTM is the first demonstration of a functionless carbamate. In light of this finding, we speculate that non-functionality is a possible feature of the many newly identified carbamate PTMs.

## 1. Introduction

Carbon dioxide is essential for life. It is at the beginning of every life process as a substrate of photosynthesis or chemosynthesis. It is at the end of every life process as the product of post-mortem decay. Therefore, it is not surprising that this gas regulates such diverse processes as cellular chemical reactions, transport, maintenance of the cellular environment, and behaviour [1]. CO_2_ is a strategically important research target for sustainable biotechnology, public health, and crop responses to environmental change. Tackling the research challenges linked to CO_2_ requires knowledge of the mechanisms for sensing, transporting, and adapting to alterations in CO_2_ partial pressure [2].

We have a growing knowledge base of signalling pathways that respond to altered CO_2_. Examples of such pathways include AMPK and Na^+^/K^+^-ATPase in mammals [3], NF-κB in mammals and *Drosophila* [4,5], the calpain/caspase-7/RhoA pathway in mammals [6], Ca^2+^ signalling in mammals [7], Gr21a/Gr63a in insects [8,9], GC-D+ neurons in rodents [10], and SLAC1/protein kinase/ABA-dependent pathways in *Arabidopsis* [11,12]. However, typically, we have less evidence for the sensing mechanisms that respond directly to the gas. Signalling molecules that respond directly to CO_2_ (or to the bicarbonate ion that is in a pH-dependent equilibrium) include the Class III nucleotidylyl cyclases of animals, fungi, and prokaryotes [2], a subset of connexins (typified by Cx26) in mammals [13], receptor protein tyrosine phosphatase *γ* of mammals [14], ubiquitin in mammals and *Arabidopsis* [4,15], PII of *Synechocystis* sp. PCC 6803 [16], and PP2C phosphatases of plants and fungi [17].

There appears to be a diversity in the mechanism by which a sensor responds to inorganic carbon. The mammalian soluble adenylyl cyclase directly responds to HCO_3_^−^ bound at the active site [18]. On the other hand, some nucleotidylyl cyclases respond directly to CO_2,_ suggesting a yet uncharacterised additional mechanism [19,20]. CO_2_ interacts directly with the PP2C phosphatases of plants and fungi via phase separation of their intrinsically disordered region [17]. The apparent majority mechanism to date is the carbamate post-translational modification (PTM) on neutral *N*-terminal α-amino- or lysine ε-amino groups [21]. The carbamate PTM forms through the nucleophilic attack of the partially positively charged carbon atom of CO_2_ by a lysine ε-amino group in its deprotonated state (Appendix A). The p*K*_aH_ of the ε-amino group for lysine in free solution (where p*K*_aH_ is defined as the pKa of the conjugate acid used to determine amine basicity) is 10.79. Therefore, lysine ε-amino group carbamylation depends on a locally privileged physicochemical environment that lowers p*K*_aH_ and enables carbamate formation. Unfortunately, such carbamates are in equilibrium with the atmosphere. So, for decades, the only known carbamates were those fortuitously identified in crystal structures, often stabilised by a local metal ion, and thus featuring a very low *k*_off_ [21].

Identifying carbamate PTMs that rapidly equilibrate with the environment and can have a role in CO_2_ sensing required new technologies to identify these ephemeral protein modifications. First, Linthwaite and co-workers used the triethyloxonium ion to ethylate carbamate PTMs, making them amenable to identification by MS/MS [22] (Appendix A). Subsequently, King and co-workers exploited the protection afforded by carbamates for electrophilic modification to identify CO_2_-binding sites [16]. These two technologies have demonstrated that the carbamate PTM is far more widespread than previously suspected. The published experimental data from these methods suggests extreme boundaries of 0.2–20% carbamylated proteins in a proteome (from data within the references).

The number of carbamate PTMs being identified poses a critical conceptual problem around the function of these PTMs in biology. We envisage one of three possibilities to be true. First, all carbamate PTMs function in CO_2_ sensing and signalling. Second, no carbamate PTMs function in CO_2_ sensing and signalling. Third, some but not all carbamate PTMs function in CO_2_ sensing and signalling. We can rule out the second possibility as there are now ample examples of functional carbamates in CO_2_ sensing and adaptive responses, as detailed above. Differentiating between the remaining two options is more problematic. This study aims to distinguish between the likelihood that all OR some carbamate PTMs function in CO_2_ sensing. We eliminated the former possibility by identifying a carbamate PTM that does not affect protein function. Therefore, we propose that functional CO_2_ sensing via the carbamate PTM arises through the cell exploiting a subset of non-specific electrophilic interactions between protein and CO_2_.

## 2. Results

Soluble *Arabidopsis* and mammalian protein lysates were equilibrated with 20 mM CO_2_/HCO_3_^−^ at pH 7.4, and TEO was added to trap carbamate PTMs by converting the carbamate PTM into a carboxyethyl group. The trapping reaction mixtures were digested with trypsin, and samples were analysed by ESI-MS/MS (Electrospray Ionisation Tandem Mass Spectrometry). The data were interrogated for variable PTMs on lysine with mass shifts of 72.0211 Da (trapped carbamate) and 28.0313 Da (*O*-ethylation on glutamate and aspartate side chains). Lysine carbamylation sites were identified on open reading frames UnitProt ID P39207 (NDKA_ARATH MS/MS peptide amino acids 1–15 MEQTFIMIKPDGVQR, proposed carbamylation on K9) and UniProt ID P15531 (NDKA_HUMAN MS/MS peptide amino acids 7–18 TFIAIKPDGVQR, proposed carbamylation on K12). The carbamylated lysines are conserved at the *N*-termini of their respective proteins (Appendix A). The proteins encode a nucleoside diphosphate kinase (NDK). Nucleoside diphosphate kinases NDK are highly conserved, multi-functional kinase enzymes [23]. They are present in all kingdoms and typically found in small, closely related families within organisms [24]. NDKs synthesise nucleoside triphosphates other than ATP. The ATP *γ*-phosphate is transferred to a nucleoside diphosphate β-phosphate via a ping-pong mechanism [25] (Appendix A). In vivo, ATP is the presumed donor nucleotide and guanosine/cytosine/uracil 5′-diphosphates (GDP/CDP/UDP) are acceptor nucleotides. K12 of human NDK-A (and presumably K9 of *Arabidopsis* NDK1 (*At*NDK1)) interacts with inorganic phosphate and a ribose hydroxyl group in the active site (Appendix A) [26]. Therefore, we initially investigated *At*NDK1 as a potential target that might reveal how CO_2_ might regulate nucleotide metabolism.

We produced *At*NDK1 as a recombinant protein as a fusion with glutathione-S-transferase (GST) (Appendix A) that was subsequently removed by proteolytic cleavage (Appendix A). ^13^C NMR is a spectroscopic technique used to analyse the carbon atom environment within a molecule. 1D ^13^C NMR can report the formation of the carbamate PTM on protein [27,28]. Protein carbonyl resonances are located in the 170–185 ppm region, whereas resonances associated with the carbamate PTM are found in the 163–166 ppm region due to the higher shielding of the carbon atom because of the delocalised nature of the electron density between the heteroatoms of the carbamate PTM.

^13^C NMR spectra supported the formation of at least one carbamate PTM on *At*NDK1 (Figure 1A). From the observed chemical shifts, the total inorganic carbon (CO_2_ + HCO_3_ = C*i*) control (green) spectrum shows a strong HCO_3_^−^ resonance at 161 ppm, while the *At*NDK1 control spectrum (red) shows only weak protein carbonyl resonances in the 171–182 ppm region and are observed in the absence of any additional C*_i_* above that found in solution in equilibrium with the atmosphere. No resonances associated with the carbamate PTM are observed in the 163–166 ppm region. The *At*NDK1 + C*_i_* spectrum (blue) has an additional peak within the region associated with the carbamate PTM at the 164.3 ppm chemical shift. The absence of this peak in the control spectra is evidence that it is derived from an interaction between *At*NDK1 and CO_2_. Furthermore, the spectra were normalised based on the intensity of the protein carbonyl peaks (171–182 ppm). Therefore, the resonance at 164.3 ppm is not due to a difference in the concentration of the *At*NDK1 samples. The observed resonance at 164.3 ppm demonstrates that *At*NDK1 binds CO_2_ in its native state.

Following the observation of a carbamate PTM in its native form on *At*NDK1 by ^13^C NMR, we used TEO trapping with ESI-MS/MS (TEO-MS/MS) as an ortholog approach to identify the specific site of carbamate formation and confirm that it matched that identified in the soluble *Arabidopsis* lysate.

A TEO-MS/MS trapping experiment was carried out on recombinant *At*NDK1 at 10 μM CO_2_ at pH 7.4, a concentration that resembles that found in intact leaves [29]. As for the soluble lysate, a carbamate modification was observed on *At*NDK1 K9 (Figure 1B). A carboxyethyl modification was observed on an internal lysine residue in the fragmented peptide, exhibiting a so-called missed cleavage. The missed cleavage occurs because lysine carboxyethylation removes the lysine cationic charge essential for trypsin’s cleavage site recognition. This observation supports identifying a carbamate PTM on *At*NDK1 K9 as a missed cleavage is an otherwise rare event. As an additional control, the TEO-MS/MS experiment was repeated using 10 μM ^13^CO_2_ at pH 7.4. The carboxyethylation modification shows a 73.0211 Da m/z shift due to the additional +1 Da m/z of the ^13^C isotope. Peptides containing a ^13^C-carboxyethylated lysine were observed, validating the prediction of a carbamate PTM at *At*NDK1 K9 (Figure 1C).

We produced a series of mutant proteins to investigate the influence of the K9 carbamate on *At*NDK1 function (Appendix A). The K9A mutant removes the K9 sidechain. We hypothesised that mutation of K9 to glutamate (K9E) would represent the local charge state of a carbamate at 100% occupancy. A mutation of K9 to arginine (K9R) was made to retain an amine in the side chain, but with a significantly higher p*K*_aH_ and thus unable to interact with CO_2_. A TEO-MS/MS trapping experiment was performed using *At*NDK1-K9R and supraphysical (1.47 mM) CO_2_. No arginine carboxyethylation modification was observed at R9, validating using this mutation.

To understand the potential impact of CO_2_ on *At*NDK1 via carbamylation of K9, we first characterized the wild-type and mutant proteins to understand the role of K9 in catalysis. Analytical size exclusion chromatography is a technique used to approximate a protein’s Stokes’ radius. Protein standards were resolved, and their retention volumes were used to generate a linear standard curve (Appendix A). The size and oligomeric state of the WT and mutant *At*NDK1 complexes were approximated based on the respective proteins’ retention volumes and the equation of the standard curve.

The elution profile of WT *At*NDK1 comprised a single peak corresponding to a predicted MW of 86 kDa (Figure 2). The ratio of this value relative to the monomeric mass is 5.2:1. Considering that the NDK quaternary structure is formed from minimal dimeric structural units [30], this value correlates to a hexameric quaternary structure consistent with the reported oligomeric state of eukaryotic NDKs [31]. The value was slightly lower than the anticipated 6:1 ratio but may be due to a discrepancy in the Stokes radius of the *At*NDK1 hexamer relative to the protein standards [32]. The elution profiles of all mutant *At*NDK1s consisted of two peaks: the putative hexameric peak and a second peak, which correlated to a molecular mass of 33 kDa (Table 1). The molecular mass of the second peak was consistent with dimeric *At*NDK1. Therefore, the stability of the *At*NDK1 hexamer was dependent on K9 and decreased as the lysine residue was substituted with a more structurally and chemically different residue.

We performed thermal shift assays as an orthologous technique to confirm the role of K9 in protein stability. Wild-type protein had a melt temperature of 63.25 *±* 0.04 *°*C. The most significant decrease in melt temperature was observed for *At*NDK1-K9E (17.30 *°*C), followed by *At*NDK1-K9A (13.35 *°*C) and *At*NDK1-K9R (11.85 *°*C), reflecting the order of the most destabilising mutation to the least destabilising mutation observed by analytical size exclusion chromatography. The destabilising effect of the K9E mutation, in particular, supported the hypothesis that introducing a similar anionic group at K9 through carbamylation would negatively impact protein function.

We assessed the biochemical parameters for *At*NDK1. Michaelis–Menten kinetics were determined for GTP by fixing [ADP] at 35 μM and varying [GTP] between 0*–*100 μM (Appendix A). The Michaelis–Menten curve gave a K_GTP_ of 22.4 μM and a V_max_ of 21.7 μmol min^−1^ mg^−1^. Michaelis–Menten kinetics were determined for ADP by fixing [GTP] at 50 μM and varying [ADP] between 0*–*100 μM (Appendix A). The data best fit a substrate inhibition model with a K_ADP_ of 24.6 μM, a V_max_ of 25.7 μmol min^−1^ mg^−1^ and a K_i_ of 132 μM. Based on these findings, 50 μM GTP and 35 μM ADP were chosen as standard assay substrate concentrations. Due to the nature of the double displacement reaction, one substrate is always limiting with respect to the other. As ADP substrate inhibition was observed below a [GTP]:[ADP] ratio of 1, an approximate ratio of 1.5:1 was deemed appropriate. Having established these concentrations, a substrate inhibition plot of *At*NDK1 activity against Mg^2+^ concentration was used to determine an appropriate assay concentration (Appendix A). *At*NDK1 activity was inhibited above 2 mM Mg^2+^ (K_i_ = 3.65 mM), potentially reflecting GTP or ADP sequestered in GTP/ADP-(Mg^2+^)_2_ complexes or protein-Mg^2+^ electrostatic interactions. Finally, a time course of the reaction demonstrated that ATP production was approximately linear over the first eight minutes (Appendix A). ADP conversion did not exceed 10% over this period; thus, the observed rate approximated the initial rate well. We subsequently determined the activity of the K9A, K9R, and K9E mutant proteins compared to *At*NDK1 wild-type (Figure 3A). A significantly reduced activity was observed for all variant proteins, consistent with the impact of the mutations on quaternary structure (Table 2).

Having established that altered K9 sidechain reactivity severely impacts catalysis, we investigated the influence of CO_2,_ which introduces an anionic group to K9 through carbamylation. Surprisingly, increasing CO_2_ over atmospheric partial pressures did not impact catalysis (Figure 3B). Two potential reasons might explain this observation, based on the hypothesis that K9 carbamylation inhibited nucleotide binding and catalysis. First, K9 carbamate occupancy was saturated at atmospheric CO_2_; therefore, changes in catalysis above this CO_2_ partial pressure would be negligible. Second, K9 carbamate occupancy was not high enough relative to nucleotide concentration, even at over 100 μM CO_2_; thus, no change in activity was observed.

To test the first hypothesis, we compared *At*NDK1 activity at 20 μM CO_2_ against a sample from which all CO_2_ had been removed by preincubating protein and assay solutions in an anaerobic chamber. We confirmed that CO_2_ was below detectable levels by confirming that no carbamates were formed on *At*NDK1 K9 by TEO-MS/MS in the absence of CO_2_ while they were still observed in samples maintained at 20 μM CO_2_. No difference in *At*NDK1 activity was observed between assays performed at 20 μM CO_2_ or in the absence of detectable CO_2_ (Figure 4A). We devised alternative assay conditions to address the second hypothesis that K9 carbamate occupation was not high enough at 100 μM CO_2_ (relative to nucleotide concentration) to observe an effect on *At*NDK1 activity. *At*NDK1 activity was measured with a decreasing concentration of total nucleotide while maintaining a constant [GTP]:[ADP] to determine the lowest nucleotide concentration at which activity was still observable. *At*NDK1 activity was still observable at 150 nM [GTP]*/*100 nM [ADP]. However, concentrations of 1.5 μM [GTP]*/*1 μM [ADP] were chosen for subsequent assays to maintain a sufficient signal-to-noise ratio for meaningful quantification. Assay [CO_2_] was increased to a supraphysiological 1 mM to increase the likelihood of observing a hypothesised K9 carbamate-mediated competitive inhibition of *At*NDK1 activity. No difference in *At*NDK1 activity was observed between assays performed at 1 mM CO_2_ or atmospheric CO_2_ at reduced nucleotide concentrations (Figure 4B).

While NDKs display clear substrate preferences for specific nucleotides [33], whether the NDK active site discriminates between NTPs and NDPs is unclear. This discrimination may not be necessary in vivo due to the difference in the donor ATP to acceptor NDP concentration [34]. We hypothesised that carbamate formation may have differential effects on nucleoside di- and triphosphates binding due to the putative transition state interaction of the K9 side chain with the *γ*-phosphate of ATP [35]. Assays at a fixed ratio of [GTP]:[ADP] may not capture such an effect if only observed when one substrate concentration is rate-limiting. A Michaelis–Menten plot at low nucleotide concentrations might address this issue, as competitive inhibition of one substrate would be reflected in the inhibition pattern observed. For example, when varying [GTP] with fixed [ADP], inhibition of GTP and ADP binding would be observed as competitive and uncompetitive, respectively. Therefore, Michaelis–Menten kinetics were determined with respect to variable [GTP] and fixed [ADP] (1 μM) at 0 mM and supraphysiological 1 mM CO_2_. (Figure 4C). The enzyme displayed almost identical K_GTP_ values under both conditions: 1.06 μM (0 mM CO_2_) and 1.02 μM (1 mM CO_2_), and no clear competitive or uncompetitive inhibition pattern was observed. Therefore, K9 carbamylation does not selectively affect the binding of one substrate.

Having demonstrated that (i) altered K9 sidechain properties are detrimental to protein structure and catalysis and that (ii) a carbamate PTM forms on K9, the failure of CO_2_ to affect *At*NDK1 activity is most likely explained if nucleotide binding competes with the freely reversible carbamate for binding to the K9 *ε*-amino group. A competitive carbamate trapping TEO-MS/MS experiment was designed to investigate the occupation of the *At*NDK1 K9 carbamate PTM in the presence of nucleotide. Due to the rapid formation of the phospho-histidine intermediate in the presence of a nucleoside triphosphate, a non-hydrolysable GTP analogue, GTP*γ*S, was chosen as a suitable competing nucleotide. Schaertl et al. demonstrated that donor ATP*γ*S reduced NDK activity a thousand-fold and that the *γ*-thio-phosphoryl transfer step was rate limiting [36]. This inhibition was attributed to the electronegativity difference between the oxygen and the sulphur of the *γ*-thio-phosphoryl group, which reduces the partial positive charge on the electrophilic phosphoryl centre, making it less susceptible to nucleophilic attack. The inhibitory effect of [GTP*γ*S] on *At*NDK1 was initially determined by measuring *At*NDK1 activity under standard assay conditions following incubation with increasing [GTP*γ*S] (Figure 5A). *At*NDK1 activity decreased with increasing [GTP*γ*S] due to competitive inhibition at the active site. Under these conditions, the IC_50_ of GTP*γ*S was 0.6805 μM. *At*NDK1 was incubated with 50 μM CO_2_ to allow carbamate formation to occur. Before initiating the TEO trapping reaction, equal volumes of GTP*γ*S (200 μM final concentration) or H_2_O were added to the GTP*γ*S(+) and GTP*γ*S(−) samples, respectively. The carboxyethylated K9 residue (IMIKPDGVQR) was observed in both samples. The relative change in the peak area of the carboxyethylated K9 peptide (across the two nucleotide conditions) was plotted against the mean change in the relative peak area of all other peptides common to both samples (Figure 5B). The peak area ratio for the carboxyethylated K9 peptide was 1.47, indicating the relative abundance of this peptide increased in the absence of GTP*γ*S. The mean ratio of all other non-carboxyethylated peptides was 0.88 ± 0.17 (95% C.I.), indicating the relative abundance of all non-carboxyethylated peptides was unaffected by GTP*γ*S. A Grubb’s Test confirmed that the value for the carboxyethylated K9-containing peptide was a significant outlier. Therefore, the inability of the K9 carbamate to influence *At*NDK1 catalysis is likely explained.

## 3. Discussion

George Lorimer suggested in 1983 that the carbamate modification might represent a widespread mechanism for a biochemical control system for altered CO_2_ levels [28]. The technical considerations identified by Lorimer that hindered the characterization of such a control system have been overcome, first by TEO-trapping [22] and, more recently, by carbamate protection of lysine side chains from homocitrullination [16]. The result is that many carbamate PTMs have been described (see Section 1). Identifying the function(s) of these ephemeral modifications at many proteome sites is challenging. However, it is not necessarily the case that the modifications need to have any function at all. We identified K9 of *At*NDK1 as a carbamylated site that has no impact on enzyme biochemistry. It is, of course, impossible to prove the negative. Yet, as far as possible, assay conditions that replicate in vivo demonstrate no impact of CO_2_ on the enzyme despite the significant influence of altered K9 side chain reactivity on protein structure and function. An unknown additional function for K9 carbamylation in vivo may exist. However, the considerable impact of K9 mutation on catalysis would make such an experiment technically unfeasible.

CO_2_ is an electrophile, and the observation that its interactions with a subset of dissociated lysines are widespread in biology is not surprising. A database of experimentally determined sidechain p*K*_a_ values provides 135 values for lysine (ε-amino group p*K*_a_ of free amino acid = 10.79), of which 14 values are <10 [37]. This observation supports a hypothesis that CO_2_-lysine interactions will be widespread in a proteome. For example, approximately 50% of inorganic carbon in the photosynthetic protozoan *Euglena gracilis* was hypothesized to be in the form of protein carbamate [38] and a significant amount of CO_2_ is protein-bound in *Synechocystis* and *Arabidopsis*, even when allowing for CO_2_ bound to Rubisco [22,39]. Our observation of a non-functional carbamate PTM offers insight into a potential interpretation. We propose that electrophilic interactions between CO_2_ and specific target lysines are widespread in biology. Further, we speculate that while an unknown percentage of those sites are non-functional, a subset has been co-opted and has a functional role in cellular responses to CO_2_. The challenge for the future is two-fold. First, it is essential to identify the full complement of CO_2_ binding sites in a proteome. Second, it will require the development of high throughput analytical methods to determine which of those sites have functional consequences for CO_2_ sensing.

## 4. Materials and Methods

### 4.1. AtNDK1 Expression and Purification 

Wild-type and mutant *Arabidopsis thaliana* NDK1 (Uniprot P39207) open reading frames were cloned into the BamHI and EcoRI sites of pGEX6P-1 in frame with the *N*-terminal glutathione-S-transferase (GST) protein by commercial gene synthesis (Genscript, Piscataway, NJ, USA). Proteins were expressed as GST-tagged fusions in *E. coli* T7 express (NEB). *At*NDK1 was expressed at 23 °C for 18 h with 400 µM IPTG. Pelleted bacteria were resuspended in equilibration buffer (50 mM Tris-HCl, 150 mM NaCl, pH 8.0) before being lysed by sonication (180 s, on ice) and centrifuged (40,000× *g*, 30 min, 4 °C). The clarified lysate was loaded onto a GSTrap™ column (1 mL, Cytiva, Lane Cove West*,* Australia), which was subsequently washed with at least ten CVs of equilibration buffer until A_280nm_ reached the baseline of the equilibration buffer. The protein was then eluted with at least five CVs of freshly prepared elution buffer (50 mM Tris-HCl, 150 mM NaCl, 10 mM reduced L-glutathione, pH 8.0) at a reduced flow rate (0.3 mL min^−1^). Fractions were analysed by SDS-PAGE, and *At*NDK1 containing fractions were dialysed into protease cleavage buffer (50 mM Tris-HCl, 150 mM NaCl, 1 mM DTT, pH 7.5) for 16 h at 4 °C, with two buffer changes. Buffer changes were essential to remove as much residual reduced L-glutathione as possible. PreScission Protease™ (1U/100 μg fusion protein) (Genscript) was added to the dialysed protein and incubated for 16 h at 4 °C. Following cleavage, the sample was dialysed into the equilibration buffer and purified with glutathione agarose as above. Cleaved *At*NDK1 was collected in the flowthrough, and the purity was analysed by SDS-PAGE. Typically, the flowthrough contained some unbound GST tag, and, thus, the sample was dialysed into size exclusion chromatography buffer (50 mM Tris-HCl pH 7.5, 200 mM NaCl, 1 mM DTT, 15% glycerol) for 16 h at 4 °C. *At*NDK1 was further purified via size exclusion chromatography using a 16/600 Superdex 200 pg column (GE Lifesciences, Marlborough, MA, USA) and an ÄKTA Start/Pure chromatography system (GE Lifesciences). Eluted fractions were analysed using SDS-PAGE, and those containing pure *At*NDK1 were combined, concentrated, and stored at −80 °C.

### 4.2. Protein CO_2_ Trapping 

Protein (50–500 µg) was diluted into 2.5 mL trapping buffer (100 mM NaH_2_PO_4_/Na_2_HPO_4_, 100 mM NaCl, pH 7.4) for CO_2_ TEO-MS/MS trapping experiments. NaHCO_3_ dissolved in trapping buffer (0.5 mL) was added to a final concentration specified for the reaction. The required NaHCO_3_ concentration was determined based on the desired CO_2_ concentration and the solution pH using the Henderson–Hasselbach equation. This solution was added to a potentiometric titrator (902 Titrando; Metrohm, Gladesville, Australia) and incubated at 25 °C with stirring for 5 min. A freshly made solution of triethyloxonium (TEO) tetrafluoroborate (240 mg, 1.47 mmol) was prepared in trapping buffer (1 mL) and added dropwise to the solution with constant stirring. The pH was maintained at the desired set point via the automated addition of NaOH (1 M), and the reaction was left stirring for 1 h to ensure complete hydrolysis of the TEO. The trapped solution was then dialysed into dH_2_O (4 °C, 16 h) with two buffer changes to ensure the removal of all hydrolysed TEO. The dialysed sample was dried at room temperature using a centrifugal vacuum concentrator (Eppendorf, Hamburg, Germany).

For *At*NDK1 trapping experiments run under CO_2_-free conditions, the reaction was carried out in a small glass vial in an anaerobic chamber (BelleTechnology, Weymouth, UK) without using the potentiometric titrator. Trapping buffer (200 mM NaH_2_PO_4_/Na_2_HPO_4_, pH 8, 100 mM NaCl) was thoroughly sparged with N_2_ before transferring into the CO_2_ free environment. *At*NDK1 (50 μg) was diluted into trapping buffer before the addition of NaCl (20 mM final concentration) to give a CO_2_(−) sample. The CO_2_(+) sample was removed from the chamber, and NaHCO_3_ was added. Samples were incubated (5 min, RT) before gradually adding the reaction mixture to solid TEO (80 mg) in 200 μL aliquots. Following the final addition, the reaction was further incubated (1 h, RT) to allow complete hydrolysis of TEO. Reactions were then dialysed and dried as outlined above.

For CO_2_-nucleotide competitive carbamate trapping assays, *At*NDK1 (50 μg total, 500 nM) was prepared in trapping buffer (100 mM NaH_2_PO_4_/NaH_2_PO_4_, pH 7.5, 100 mM NaCl, 2 mM MgCl_2_, 1 mM NaHCO_3_). The reaction mixture was incubated (5 min, RT) to allow carbamate formation before the addition of the competing nucleotide (2 mM adenosine diphosphate) (Sigma Aldrich, St. Louis, MO, USA)/0.2 mM guanosine 5′-O-[*γ*-thio] triphosphate (GTP*γ*S) (Sigma Aldrich) or mQ H_2_O for control reactions. Reaction mixtures were further incubated (5 min, RT). The trapping reaction was then carried out as outlined above.

### 4.3. Mass Spectrometry 

S-Trap™ (Protifi, Fairport, NY, USA) Mini digestion was performed using LC-MS grade reagents and according to the manufacturer’s instructions with modifications. Dried protein (~100–300 µg) was resuspended in 1× SDS Lysis buffer (5% (*v/v*) SDS, 50 mM triethylammonium bicarbonate (TEAB) pH 8.5, 50 µL). Samples were diluted in an equal volume 2× SDS Lysis Buffer (10% (*w/v*) SDS, 100 mM TEAB pH 8.5). DTT (20 mM) was added, and the sample was heated (10 min, 95 °C) to reduce disulfide bonds. Iodoacetamide (40 mM) was added, and the sample was incubated in the dark (30 min) to alkylate fully. Phosphoric acid (~1.2%) was added to the supernatant before the addition of S-Trap Binding Buffer (90% (*v/v*) methanol, 100 mM TEAB pH 7.55, 6× total sample volume). The resulting colloidal solution was loaded onto the S-Trap Mini spin column and centrifuged (4000× *g*, 30 s) to bind the protein to the S-trap. The column was then washed with S-Trap binding buffer (400 µL) and centrifuged (4000× *g*, 30 s). This washing was carried out five times before the S-Trap Mini spin column was transferred to a clean collection tube. A freshly made digestion solution prepared of Trypsin Gold (Promega, Madison, WI, USA, Mass Spectrometry Grade) (1:20 (*w/w*) Trypsin Gold: Sample) in digestion buffer (50 mM TEAB, pH 8.5, 125 µL total volume), which was added to the column. The column was briefly centrifuged (4000× *g*, 2 s), and any flowthrough was reloaded onto the column and then incubated (37 °C, 16 h). Peptides were eluted from the column by adding three elution buffers, with each addition followed by a centrifugation step (1000× *g*, 60 s). Elution Buffer 1 (50 mM TEAB, pH 8.5, 80 µL) was used to elute most of the aqueous peptides, followed by Elution Buffer 2 (0.2% (*v/v*) formic acid, 80 µL). Finally, Elution Buffer 3 (50% (*v/v*) acetonitrile (ACN), 0.2% (*v/v*) formic acid, 80 µL) was used to elute hydrophobic peptides. Eluted peptides were combined and dried at room temperature using a vacuum centrifuge.

The digested peptides were desalted on a C18 column and analysed by ESI-MS/MS on an LTQ Orbitrap XL mass spectrometer (ThermoFisher, Waltham, MA, USA) coupled to an Ultimate 3000 nano-HPLC instrument. Peptides eluted from the LC gradient were injected online to the mass spectrometer (lock mass enabled, mass range 400–1800 Da, resolution 60,000 at 400 Da, 10 MS/MS spectra per cycle, collision-induced dissociation (CID) at 35% normalised CE, rejection of singly charged ions).

The LC-MS/MS raw data files (.wiff) were converted into .mgf or .mzXML files using the freeware MSConvert (ProteoWizard, Palo Alto, CA, USA) and analysed using PEAKS Studio 10.5 software. An error tolerance of 15.0 ppm for the precursor mass using the monoisotopic mass and 0.2 Da for the fragment ion was used. Tryptic digests were selected using a semispecific digest mode and a maximum of three missed cleavages per peptide. Protein modifications used were fixed (57.0215 Da@C) or variable (15.9949 Da@M, 42.0106 Da@*N*-term/K, 72.0211 Da@*N-*term/K, 73.0211 Da@*N-*term/K, 28.0313 Da@*N-*term/D/E/K).

### 4.4. ^13^C-Nuclear Magnetic Resonance

Protein was exchanged into NMR sample buffer (100 mM NaH_2_PO_4_/Na_2_HPO_4_, pH 7.6, 100 mM NaCl) using a centrifugal concentrator (Vivaspin Sartorius, Goettingen, Germany). The total sample volume was 0.7 mL with protein. Samples contained 10% (*v/v*) D_2_O, and inorganic carbon was added as NaHCO_3_. ^13^C-NMR spectra were acquired with a Varian 600 MHz spectrometer equipped with an Agilent OneNMR Probe to deliver a maximum pulsed-field gradient strength of 62 G cm^−1^. A ^1^H spectrum was acquired to examine for small molecule impurities. Thirteen ^1^H experiments were recorded in 12 h, collecting 131,072 complex points. The repetition time was 6.7 s, of which 1.7 s comprised the acquisition time. The excitation pulse angle was set to 45 degrees. The strong interfering H_2_O signal was eliminated using the Robust-5 pulse sequence. Thirty-two ^13^C scans were collected, comprising 65,536 complex data points and a spectral width of 10 kHz. The repetition time was 6.3 s, of which 3.3 s comprised the acquisition time. The W5 inter-pulse delay was set to 240 µs. Rectangular 1 ms pulsed-field gradients were used in all cases with a strength of G1 = 28.3 G cm^−1^ (first pair) and G2  =  4.9 G cm^−1^ (second pair). The gradient stabilisation delay was 0.5 ms. The first pair of lock pre-focusing field gradients were separated from the first radio-frequency pulse by a 1.5 ms delay.

### 4.5. Analytical Size Exclusion Chromatography

Analytical sizing was carried out using a Superose 6 increase 10/300 GL size exclusion chromatography column (GE Lifesciences) with an ÄKTA Pure chromatography system (GE Lifesciences). A molecular weight standard curve was generated using a gel filtration calibration kit (GE Lifesciences). The column was equilibrated with size exclusion buffer (30 mM Tris pH 7.9, 200 mM NaCl, 1 mM DTT, 1 mM EDTA) and calibrated using two 100 µL injections consisting of a different combination of molecular weight proteins. The elution volumes were used to determine a standard curve plotting *K_av_* against Log_10_(M_r_). A sample (150 µL, ~1 mg mL^−1^) was prepared for each protein, of which 100 µL was injected onto the column.

### 4.6. Thermal Shift Assays

Protein (0.5 mg/mL) was prepared in 20 mM NaH_2_PO_4_/Na_2_HPO_4_, 100 mM NaCl, 2 mM MgCl_2_ pH 7.5. The protein solution was mixed with SYPRO™ orange dye (20×) by gentle inversion. This solution (10 µL/well) was transferred to a 96-well qPCR plate (ThermoFisher, Waltham, MA, USA) before the addition of further mQ H_2_O (10 µL). The plate was sealed and centrifuged (600× *g*, 2 min) to ensure mixing before being covered and incubated (4 °C, 10 min). The thermal shift assay was carried out in an RT-PCR machine and processed using the GUI-driven software NAMI (version 2.7).

### 4.7. Nucleoside Diphosphate Kinase Assay 

Nucleoside diphosphate kinase assays were performed in a reaction buffer (50 µL total volume) consisting of NaH_2_PO_4_/Na_2_HPO_4_ (50–100 mM, pH 7.6), KCl (50 mM) and MgCl_2_ (2 mM). *At*NDK1 was diluted in mQ H_2_O and added to a final concentration of 1 nM. Assays were initiated by adding a nucleotide mix of adenosine-5′-diphosphate (ADP) and guanosine-5′-triphosphate. Assays were incubated (25 °C, 6 min) before being terminated by boiling (95 °C, 5 min) or EDTA addition (10 mM final concentration). Total ATP production was measured via a Molecular Probes^TM^ (Invitrogen^TM^, Carlsbad, CA, USA) ATP determination assay in which the assay was diluted in mQ H_2_O, such that ATP production was within the linear range of the assay ATP standard curve. ATP production was subsequently used to determine the *At*NDK initial rate. Relevant controls for all experiments demonstrated that assay conditions did not interfere with the ATP determination assay. Depending on the assay, inorganic carbon was added as HCO_3_^−^/CO_2_, with [anion] maintained with supplemental Cl^−^.

### 4.8. Statistical Analysis

All data are presented as the distribution of independent data points representing independent experiments. All statistics and graphical analyses were performed using GraphPad Prism 10 (GraphPad Software, Inc., Boston, MA, USA).

## Figures and Tables

**Figure 1 ijms-25-00898-f001:**
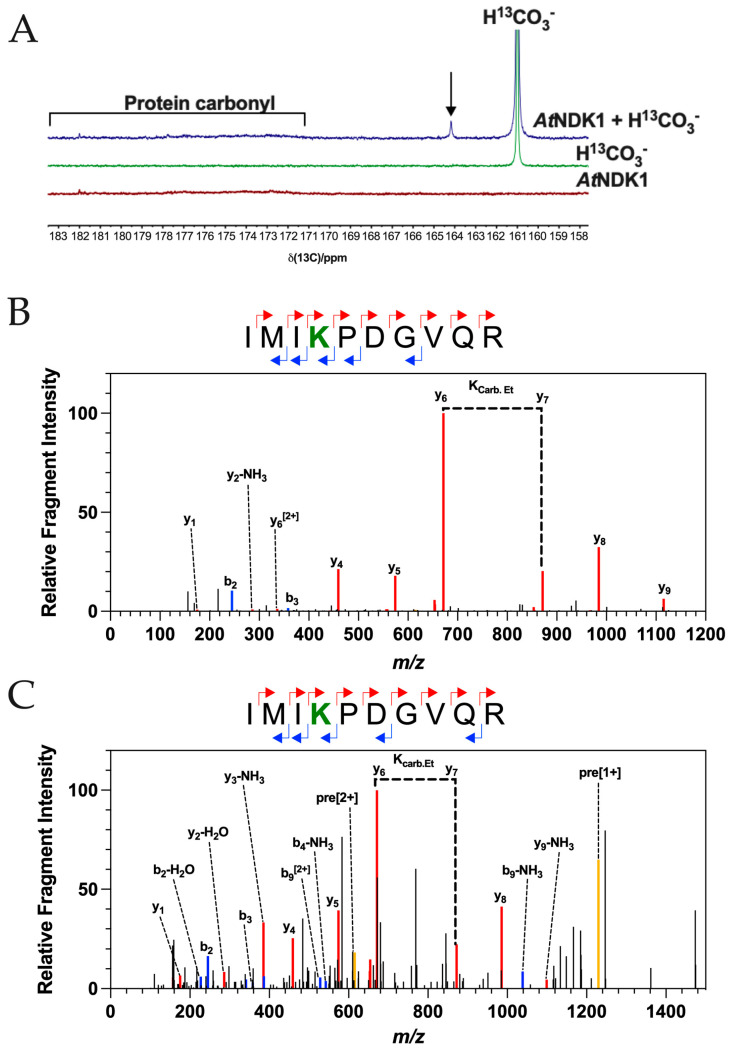
CO_2_ forms carbamates on *At*NDK1. (**A**) The 1D ^13^C–NMR spectra of 2.5 mM *At*NDK1 wild-type alone, 50 mM NaH^13^CO_3_ alone, and 2.5 mM *At*NDK1 wild-type with 50 mM NaH^13^CO_3_ are shown. The background H^13^CO_3_^−^ is observed along with the presumed carbamate (arrow) and protein carbonyl resonance. (**B**,**C**) Plots of relative fragment intensity versus mass/charge ratio (*m/z*) for fragmentation data from MS/MS identifying the carboxyethyl modification on recombinant *At*NDK1 at K9 using ^12^CO_2_ (**B**) or ^13^CO_2_ (**C**). Peptide sequences indicate predominant +1y (red) +1b (blue) ions identified by MS/MS shown in the plot. The modified residue is displayed in bold. K_carb.Et_ indicates the molecular weight difference between ions diagnostic of the modified Lys. Yellow lines indicate the precursor ion.

**Figure 2 ijms-25-00898-f002:**
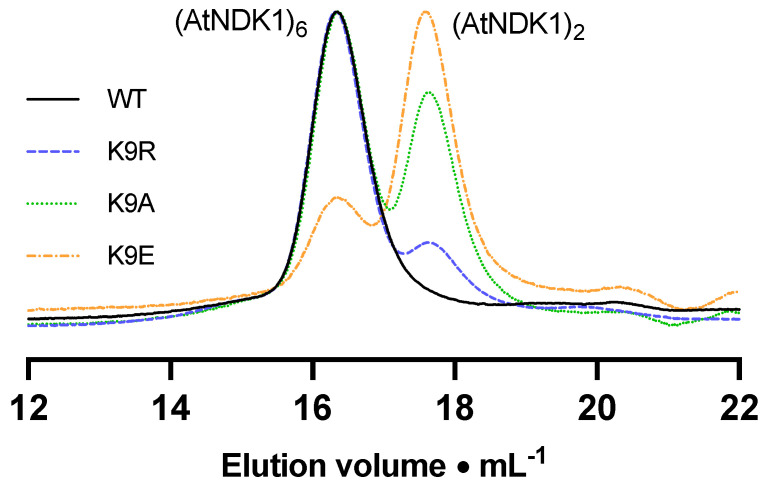
K9 point mutations alter the elution profile for *At*NDK1. A plot of normalised A_280_ vs. chromatography column retention time for *At*NDK1 WT and K9.

**Figure 3 ijms-25-00898-f003:**
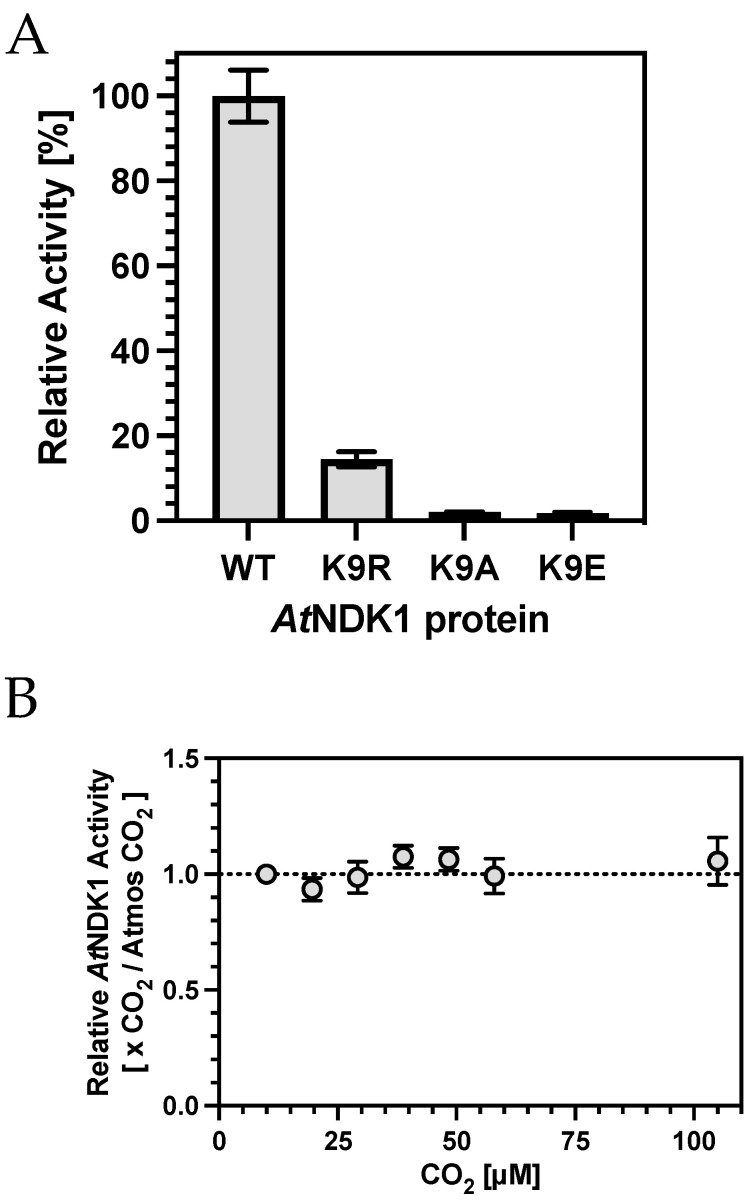
(**A**) K9 is essential for *At*NDK1 kinase activity. The relative activities of the K9 mutant proteins were determined as a percentage of wild-type (WT) activity. Bars represent mean ± S.E.M, *n* = 3. (**B**) Relative *At*NDK1 kinase activity was not affected by increasing [CO_2_]. *At*NDK1 activity was determined relative to activity observed at atmospheric [CO_2_] and plotted against [CO_2_]. Each point represents mean ± S.E.M, *n* = 4.

**Figure 4 ijms-25-00898-f004:**
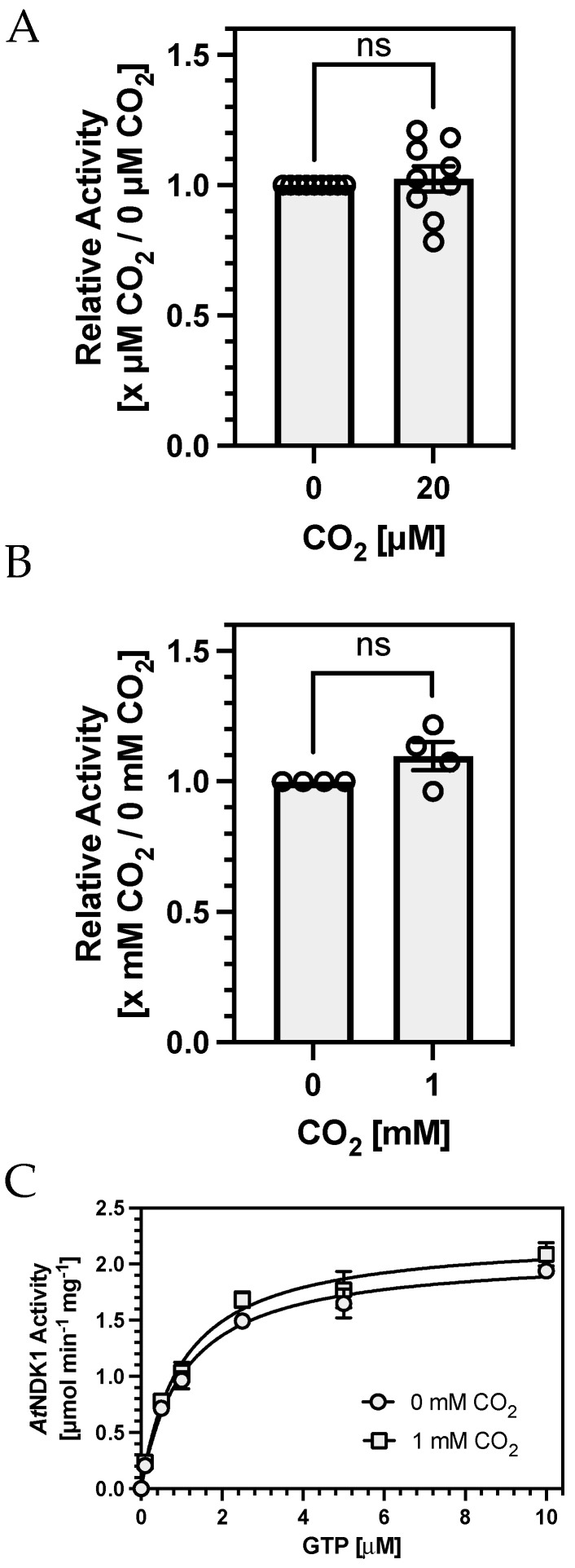
Relative *At*NDK1 kinase activity was not affected by increasing [CO_2_]. (**A**) *At*NDK1 activity at 0 μM was normalised to 1, and the activity ratio at 20 μM was determined. Bars represent the mean ± S.E.M, *n* = 9. One sample *t*-test; theoretical mean 1.000. 20 μM CO_2_, mean = 1.023 ± 0.045. *p* = 0.6423, t = 0.4826, df = 8. (**B**) *At*NDK1 activity at atmospheric [CO_2_] was normalised to 1, and the ratio of the activity at 1 mM was determined. Bars represent the mean ± S.E.M, *n* = 4. One sample t−test; theoretical mean 1.000. 1 mM CO_2_; mean = 1.097 ± 0.0464. *p* = 0.1676, t = 1.812, df = 3. (**C**) Michaelis–Menten curve (0*–*10 μM GTP) at fixed 1 μM ADP at atmospheric and 1 mM CO_2_. Each point represents the mean ± S.E.M, *n* = 5. Atmospheric CO_2_; V_max_ = 2.086 μmol min^−1^ mg^−1^, K_GTP_ = 1.06 μM. 1 mM CO_2_; V_max_ = 2.248 μmol min^−1^ mg^−1^, K_GTP_ = 1.018 μM.

**Figure 5 ijms-25-00898-f005:**
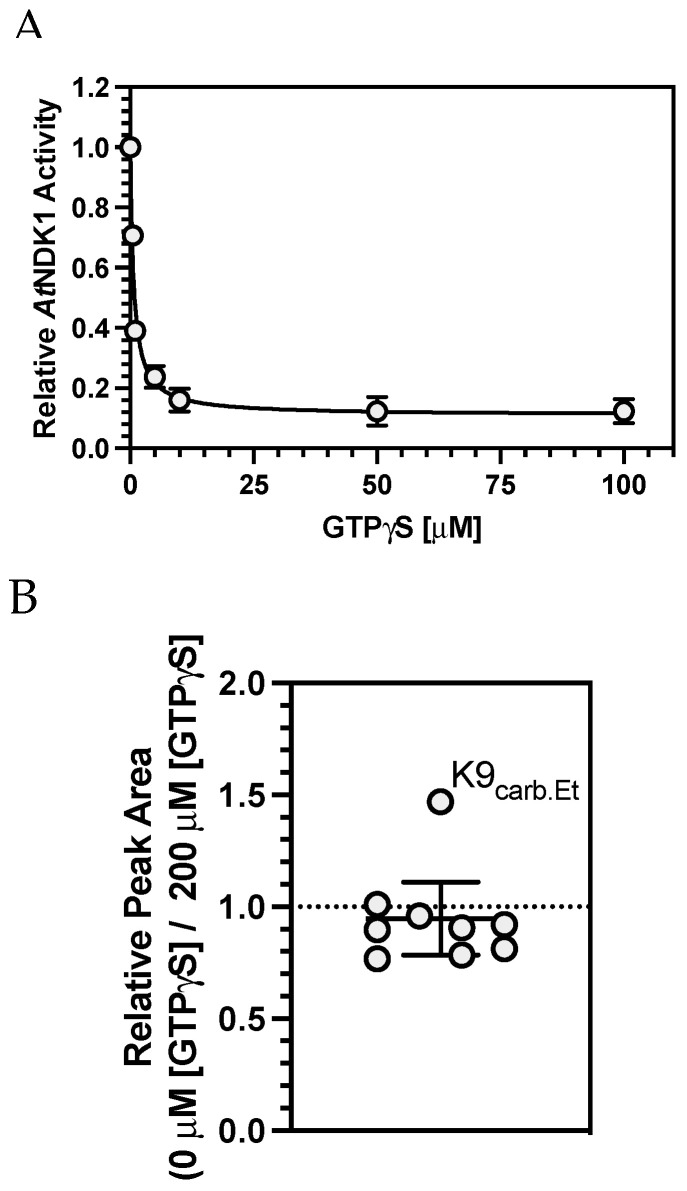
Relative *At*NDK1 carbamylation is affected by substrate. (**A**) Relative *At*NDK1 kinase activity determined as a proportion of maximal activity in the absence of GTP*γ*S plotted against increasing [GTP*γ*S]. Each point represents mean ± S.E.M, *n* = 3. (**B**) *At*NDK1 TEO-trapping with 50 μM CO_2_ and with or without 200 μM GTP*γ*S. The graph plots the ratio of peak areas for peptides in common between all experiments under the two nucleotide conditions. Each data point represents the sum of three independent replicates. The K9_carb.Et_ point is an outlier (Grubb’s Test, *p* < 0.01, Z = 2.47).

**Table 1 ijms-25-00898-t001:** Relative proportions of hexameric and dimeric *At*NDK1 complexes. The ratio of the absorbance maxima for each peak was determined relative to the normalised maxima within each chromatogram.

Protein	(*At*NDK1)_6_	(*At*NDK1)_2_
Wild-type	1.00	0.10
K9R	1.00	0.25
K9A	1.00	0.73
K9E	0.40	1.00

**Table 2 ijms-25-00898-t002:** Relative activities of *At*NDK1 proteins (±S.E.M., *n* = 3).

Protein	Relative Activity [%]
Wild-type	100.0 ± 6.1
K9R	14.4 ± 1.8
K9A	1.9 ± 0.1
K9E	1.8 ± 0.1

## Data Availability

The data presented in this study are available within the article or Appendix A.

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
