# Peer review of "A Non-Functional Carbon Dioxide-Mediated Post-Translational Modification on Nucleoside Diphosphate Kinase of Arabidopsis thaliana"

_ijms, 2024, doi:10.3390/ijms25020898_

Round 1

Reviewer 1 Report

Comments and Suggestions for Authors

This is a sound work, well made and substantiated, dealing with a very interesting subject of great biological implications. It could be well cited in the near future.

This reviewer has only very minor suggestions for improvement:

-in page 2, rows 83-85, authors should explain better that the masses of 72.0211 and 28.0313 correspond not to the whole modified residues but to the shifts in residue masses due to trapped carbamate (the former) and o-ethylation (the latter), as in fact they correctly stated at the Materials and Methods Section (rows 443-444).

-in page 3, row 104, it would be more precise and understandable to use here the term "proteolytic cleavage by PreScission ProteaseTM", or just "proteolytic cleavage", instead of "enzymatic cleavage".

-in page 4, subFig. 1A, the lettering along the abscissa axis is excessively small and should be increased in size.

-reference 37, of S. Pahari et al. 2019, a non-standard one, could be better completed by adding page number (1-7) and doi:10.1093/database/baz024

Author Response

This is a sound work, well made and substantiated, dealing with a very interesting subject of great biological implications. It could be well cited in the near future.

This reviewer has only very minor suggestions for improvement:

We thank the reviewer for their comments

-in page 2, rows 83-85, authors should explain better that the masses of 72.0211 and 28.0313 correspond not to the whole modified residues but to the shifts in residue masses due to trapped carbamate (the former) and o-ethylation (the latter), as in fact they correctly stated at the Materials and Methods Section (rows 443-444).

This change has been made. The sentence now reads:

The data were interrogated for variable PTMs on lysine with mass shifts of 72.0211 Da (trapped carbamate) and 28.0313 Da (O-ethylation on glutamate and aspartate side chains)

-in page 3, row 104, it would be more precise and understandable to use here the term "proteolytic cleavage by PreScission ProteaseTM", or just "proteolytic cleavage", instead of "enzymatic cleavage".

This change has been made. The sentence now reads: 

We produced AtNDK1 as a recombinant protein as a fusion with glutathione-S-transferase (GST) (Figure S3A) that was subsequently removed by proteolytic cleavage (Figure S3B-C). 

-in page 4, subFig. 1A, the lettering along the abscissa axis is excessively small and should be increased in size.

The requested change to the figure has been made.

-reference 37, of S. Pahari et al. 2019, a non-standard one, could be better completed by adding page number (1-7) and doi:10.1093/database/baz024

The requested change to the reference has been made.

Reviewer 2 Report

Comments and Suggestions for Authors

This is a very interesting and verywell written paper. I had red it with pleasure.

Inmy opinion three small corrections might be beneficial to the paper:

1./ there is no chemical reaction scheme given  - thus, it requires from the reader the recollection of its course;

2./ I would like to propose to widen the figures 1B and 1C (then differences will be better visualised); explaantion of the yellow line is desirable;

3./ it also would be more visible if the "K" in the sequence will be coloured.

Author Response

This is a very interesting and verywell written paper. I had red it with pleasure.

Inmy opinion three small corrections might be beneficial to the paper:

1./ there is no chemical reaction scheme given  - thus, it requires from the reader the recollection of its course;

We have added a reaction scheme for lysine carbamylation (new Figure S1A), for TEO trapping (new Figure S1B), and for NDKs (new Figure S2B).

2./ I would like to propose to widen the figures 1B and 1C (then differences will be better visualised); explaantion of the yellow line is desirable;

The requested change to the figure has been made. The yellow line has been defined in the figure 1C legend thus:

Yellow lines indicate the precursor ion.

3./ it also would be more visible if the "K" in the sequence will be coloured.

The requested change has been made.